# Performance Analysis of Quantum Key Distribution Technology for Power Business

**Bingzhen Zhao [1,2], Xiaoming Zha [1], Zhiyu Chen [3,*] , Rui Shi [3], Dong Wang [1], Tianliang Peng [4,*] and Longchuan Yan [3]**

[1]  School of Electrical Engineering and Automation, Wuhan University, Wuhan 430072, China;
    bingzhenzhao@sgec.sgcc.com.cn (B.Z.); xmzha@whu.edu.cn (X.Z.); wangdong@sgec.sgcc.com.cn (D.W.)
[2]  State Grid Electronic Commerce Co., Ltd., Beijing 100053, China
[3]  State Grid Information & Telecommunication Branch, Beijing 100761, China; rui-shi@sgcc.com.cn (R.S.);
    lchyan@sgcc.com.cn (L.Y.)
[4]  School of Information Engineering, Nanchang Institute of Technology, Nanchang 330099, China
*   Correspondence: zhiyu-chen@sgcc.com.cn (Z.C.); pengtl@nit.edu.cn (T.P.)

**Abstract:** Considering the complexity of power grid environments and the diversity of power communication transmission losses, this study proposes a quantum key distribution (QKD) network structure suitable for power business scenarios. Through simulating the power communication transmission environment, performance indicators of quantum channels and data interaction channels in power QKD systems are tested and evaluated from six aspects, such as distance loss, galloping loss, splice loss, data traffic, encryption algorithm and system stability. In the actual environment, this study combines the production business to build a QKD network suitable for power scenarios, and conducts performance analyses. The experimental results show that power QKD technologies can meet the operation index requirements of power businesses, as well as provide a reference for large-scale applications of the technology.

**Keywords:** quantum key distribution; power business; power communication; network security

## 1. Introduction

The power Internet of Things (IoT) revolves around all aspects of the power system, fully applying modern information technologies such as artificial intelligence, mobile interconnection and advanced communication technologies to achieve interconnection of all things and human–computer interaction in all aspects of the power system [1]. The ecological environment of power industry is being changed to form an interactive power grid interconnected by traditional power systems, distributed energy sources and information and communication systems. By analyzing the incidents of power system attacks in the world in recent years (e.g., man-in-the-middle attack, data exposure and code instrumentation), the continuous deterioration of the network environment and the rapid evolution of network attack technologies have further highlighted the importance of power information communication security [2–4]. With the rapid construction of power IoT, the security of data interaction between power systems depends on the adopted security protection mechanism. Although the information encryption mechanisms of different power systems are different, the commonality is that the security of the key directly affects the safe operation of the power system. Currently, there are two main ways to update the power system key: online and offline, respectively. Some power dispatching and control businesses mainly rely on offline way to update the key, which leads to the insecure and insufficient timeliness of key distribution. Most management businesses use the online key update

method; endless network attack methods make key distribution face the hidden danger of being stolen. Once a key is obtained by the third party, its attack will cause serious threat to the power system [5,6].

As a new method to ensure the secure transmission of information, QKD technology has attracted the attention of all walks of life and has successively carried out verification tests in many fields. It uses the physical characteristics of photons to solve the problem of online secure distribution of keys and realizes the anti-theft and tamper-proof of keys. QKD technology can greatly improve information security in the fields of national defense, government affairs, finance, power and energy [7–9]. The implementation of projects such as the United States defense advanced research projects agency QKD network in 2005, the European secure communication based on quantum cryptography network in 2008, and the Tokyo QKD network in Japan in 2010 marked that this technology has become one of the key technologies for development [10–12]. Since 2012, various countries have proposed plans to build QKD networks. In 2015, China launched the construction of a QKD network spanning 2000 km (Beijing–Shanghai Trunk Line), which has been piloted in banking and national defense [13]. The banking industry has carried out security transmission of business data, such as data disaster preparedness, information collection and public opinion confidentiality. China Telecom has built Shanghai Lujiazui financial QKD network, improving the protection ability of user data. In order to ensure the safe transmission of privacy data, many local governments have constructed the QKD network of e-government. However, compared with the line environment and business scenarios of power industry, the existing QKD application mode is single, and the quantum channel is dominated by buried optical fiber cable (BOFC), which is less affected by the external environment. Therefore, it is difficult to provide effective reference for special industries.

Since 90% of power communication networks are overhead lines, with long distances between stations and a large number of nodes, QKD performance is easily affected by environmental factors such as wind, rain, snow and electromagnetism [14,15]. Several researches on the application of QKD in power industry have been carried out. A QKD scheme that was independent of line polarization had achieved good results in wide area power network [16]. However, its low-key rate limited its application in multi-service scenarios. A QKD networking model was introduced, but only studies the effect of distance on QKD performance [17]. A method of power-securing events and unified meter reading businesses based on QKD was proposed [18]. It provided a secure transmission channel for unified billing information of power, water and gas, but the network was mainly based on BOFC. A multi-point power dispatching quantum key application strategy was proposed, which effectively improved the secure transmission level of the dispatching network. However, it did not take into account the actual operating environment [19,20]. Article [21] used a local search algorithm to improve the efficiency of MDI-QKD (measurement device independent-quantum key distribution protocol) and designed a quantum key application strategy in a simulation environment to improve the utility of QKD in power dispatching network.

For the practical application of QKD technology in power industry, the existing research results have not been systematically verified, and the combination with the existing power business is insufficient [22–24]. Considering the differences among the attributes of power businesses, each business has different encryption requirements and security requirements. At the same time, power dispatching and control services have extremely high requirements for communication delay and reliability. The real-time requirement of power financial data transmission is less demanded, however, it contains a lot of sensitive information, so the confidentiality requirement of information interaction can be extremely high. Not like text encrypted transmission, video conference involves a large amount of data, as well as higher real-time and synchronization requirements. In complex power environments, rigorous testing and verification are required to verify whether QKD technology can operate safely and stably. Therefore, it is necessary to carry out the performance evaluation of the power quantum secret communication system under the complex grid environment.

Based on the urgent needs of power business security, this paper simulates a variety of different power grid operating environments, and the performance of QKD technology in power business

has been verified. Meanwhile, it is verified whether the QKD technology can satisfy the business operation indicators of power dispatching, system protection, disaster recovery data and video conference. Meanwhile, in order to improve the security protection ability of power service information interaction, this paper studies the QKD networking in typical business scenarios. The fast polarization feedback technology is adopted to solve the stability problem of overhead power line quantum key encryption. Based on this, it is further combined with business attributes such as grid dispatching, power distribution automation, power information collection and video conference to build a secure and reliable communication network.

This paper is organized as follows: the principle of BB84 protocol and QKD is briefly introduced in Section 2. In Section 3, we describe the performance characteristics of power QKD. Several QKD networking schemes are proposed in Section 4. The experimental results of the power QKD system are presented and discussed in Section 5. Finally, the conclusion is summarized.

## 2. Review of QKD Technology

The development of QKD technology is based on communication theory and quantum mechanics. Through the generation, modulation and detection of quantum signals, information transmission is realized. According to the principle of indivisibility, uncertainty and no-cloning of single photons, any third-party behavior will affect the state of photon. Based on QKD protocol, the two parties negotiate and generate an absolutely secure quantum key based on the photon physical state as the information carrier. There are several types of QKD protocol, such as weak coherent pulses, true single photon signals, entangled photon signals and continuous variable signals.

The BB84 protocol proposed by Charles Bennett and Gilles Brassard in 1984 is the earliest quantum confidential communication protocol [25]. With the continuous research, a variety of QKD protocols based on the BB84 protocol have been proposed. The B92 protocol simplifies the BB84 protocol but reduces communication efficiency and practicability [26]. The six-state protocol is extended on the basis of the BB84 protocol, which raises the upper limit of the quantum error rate [27]. Limited to the existing single photon source technology, it is not possible to obtain ideal single photon. The decoy protocol was proposed to improve non-ideal single photon source BB84 protocol, which solves the problem of "photon number separation attack" [28]. At present, single photon QKD system mainly include polarization encoding and phase encoding. Taking polarization encoding as an example, the horizontal polarization state and vertical polarization state, ±45° polarization state of the photon are used for encoding. Assuming that the state of photon polarization is horizontal or −45° is defined as "0", then the description of vertical or 45° is "1". And there are corresponding measurement bases that are conjugate to each other. The sender sends the polarization state to the receiver. The receiver randomly selects one of the two conjugate groups and uses a photon polarization detector to measure the quantum polarization state. If the measurement base is the same as the transmission base, the accuracy of the polarization direction can be guaranteed. If the measurement base is not the same as the sending base, there is only a 50% probability that the correct polarization direction can be determined, and complete information cannot be accurately measured.

Figure 1 and Table 1 describe the working principle of the quantum key distribution and the formation process of the quantum key, respectively. The transmitter of QKD system includes components, such as quantum signal source, modulator and random number generator and other components. First, a random number is generated by random number generator and then single-photon transmissions with different polarization states are prepared. After the receiver receives the single photon signal, it randomly selects the basis vector for measurement, and feeds back the measured basis vector to the sender. Finally, the sender tells the receiver the location of the common basis vector, and both parties retain the measurement result of the same basis vector as the quantum key. Meanwhile, the system selects a small number of quantum keys to calculate the bit error rate to determine whether there is third party behavior. In practical applications, the classic common signal in Figure 1 can use

the same one. In this paper, we conduct research on the application of QKD technology based on polarization encoding in power industry.

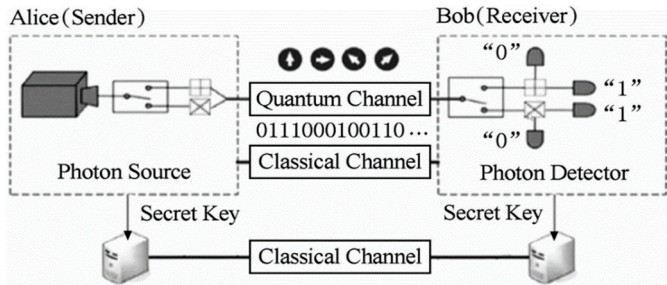

**Figure 1.** Quantum key distribution (QKD) based on polarization encoding.

**Table 1.** Quantum key generation based on the BB84 Protocol.

| QKD | Examples of QKD | | | | | | | |
|---|---|---|---|---|---|---|---|---|
| Alice's bit sequence | 0 | 1 | 0 | 0 | 1 | 1 | 0 | 1 |
| Alice's basis | + | + | × | × | + | + | × | × |
| Alice's polarization | ↑ | → | ↗ | ↗ | → | → | ↗ | ↘ |
| Bob's basis | × | + | × | + | × | + | + | × |
| Bob's measurement | ↗ or ↘ | → | ↗ | ↑ or → | ↗ or ↘ | → | ↑ or → | ↘ |
| Shared sifted key | - | 1 | 0 | - | - | 1 | - | 1 |

## 3. Performance Characteristics of Power QKD System

As the transmission of strong light signals, quantum light signals are affected by media properties such as fiber loss and fiber dispersion. At the same time, due to the diversity and complexity of the operating environment of the power network, the optical signals with stronger quantum light signals are more susceptible to environmental factors. In this section, we analyze the key factors that affect the performance of a simulated power communication QKD network.

### 3.1. Performance Index Analysis

Compared with the QKD network of other industries, the power communication network coexists with the power transmission network, which has the characteristics of wide geographical area, diverse line environment and complex network structure. Since power communication network is composed of overhead optical fiber and BOFC, the traditional QKD system is easily affected by the external wind, rain, snow and other environment factors, resulting in low generation efficiency of quantum cryptography. To evaluate the practical performance of power QKD system more effectively, this article proposes the evaluation index from the key tier and the business layer for the situations that may occur in the actual link, as shown in Figure 2. The key tier evaluation index is reflected in the factors that affect the secret key rate in the quantum channel. The business layer evaluation indicators focus on the transmission performance of the system when quantum virtual private network (QVPN) is used for encrypted transmission. In order to test the performance of QKD device and QVPN in different environments, we simulated different test environments.
Case 1: The Key Tier

(1) Distance loss test: by setting up fiber optic environments with different transmission distances, test the secret key rate of QKD devices under different distance attenuation conditions, and record the coding curve within one hour after stable coding. The actual line is BOFC (16.58 km). The simulated environment consists of four bare fibers with lengths of 10 km, 20 km, 30 km and 40 km.

(2) Galloping loss test: on the basis of distance test, the galloping simulation test environment of optical fiber composite overhead ground wire (OPGW) is established. Test the secret key rate of the

quantum device under different wind conditions within the transmission distance, and statistically determine the code curve within one hour after the code is stable. Wind level: calm, light air, light breeze, gentle breeze and moderate breeze (Table 2).

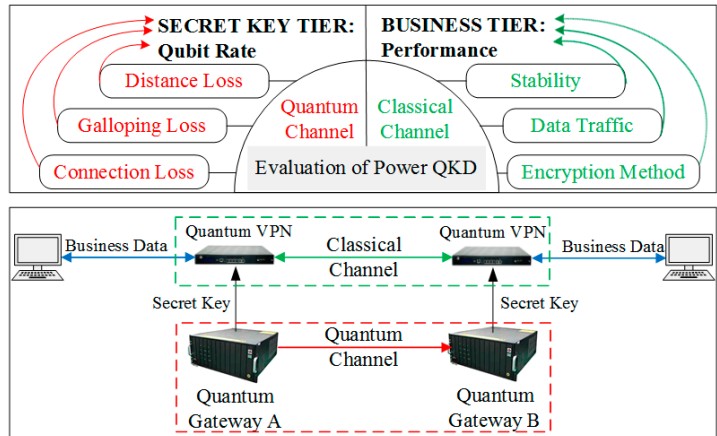

**Figure 2.** Evaluation indicators of power QKD system.

**Table 2.** Wind scale and wind speed list.

| Wind Scale | Wind Speed (m/s) |
| --- | --- |
| Calm | 0–0.2 |
| Light air | 0.3–1.5 |
| Light breeze | 1.6–3.3 |
| Gentle breeze | 3.4–5.4 |
| Moderate breeze | 5.5–7.9 |

(3) Splice loss test: in a fixed transmission distance, by inserting splice fibers with different losses, test the effect of fiber environment on the secret key rate of the quantum device under different splice losses. In this experiment, three different spliced optical fibers were used to test the analog lines.

Case 2: The Business Tier

(1) Data traffic test: by loading a network performance tester at both ends of the service, test the transmission performance parameters such as network delay, jitter, throughput and packet loss rate when the classic channel uses QVPN encrypted transmission.

(2) Encryption algorithm test: a network performance tester is used to test transport performance when QVPN uses different encryption algorithms to encrypt business data. QVPN supports three encryption methods, such as Internet key exchange (IKE), SM4/AES + quantum key.

(3) System stability test: by connecting the actual loop fiber-optic line, test the secret key rate of the quantum device in the actual transmission environment within $7 \times 24$ h.

*3.2. Simulation Environment Test*

Figure 3 shows the topology of test system. In this experiment, the quantum channel was tested using simulated fiber channels and actual fiber channel. In the simulation of the optical fiber line test, the quantum link uses different kilometers of fiber optic disks as the transmission medium. In the actual fiber-optic line test, the quantum link uses the buried optical cable in the actual environment. The hardware equipment is shown in Table 3. The following describes the use cases of this experiment.

(1) Distance loss test—tests the secret key rate under 10 km, 20 km, 30 km, 40 km and the actual link.

(2) Galloping loss test—tests the key rate of 10 km, 20 km, 30 km and 40 km under different wind scale.

(3) Splice loss test—detects the effect of splice loss on secret key rate (wavelength: 1550 nm).

(4) Encryption algorithm test—tests the impact of different encryption algorithms on secret key rate.

(5) Data traffic test—tests the impact of data traffic on the secret key rate and the maximum traffic supported by QVPN.

(6) Stability performance test—tests the secret key rate of the actual link under light breeze.

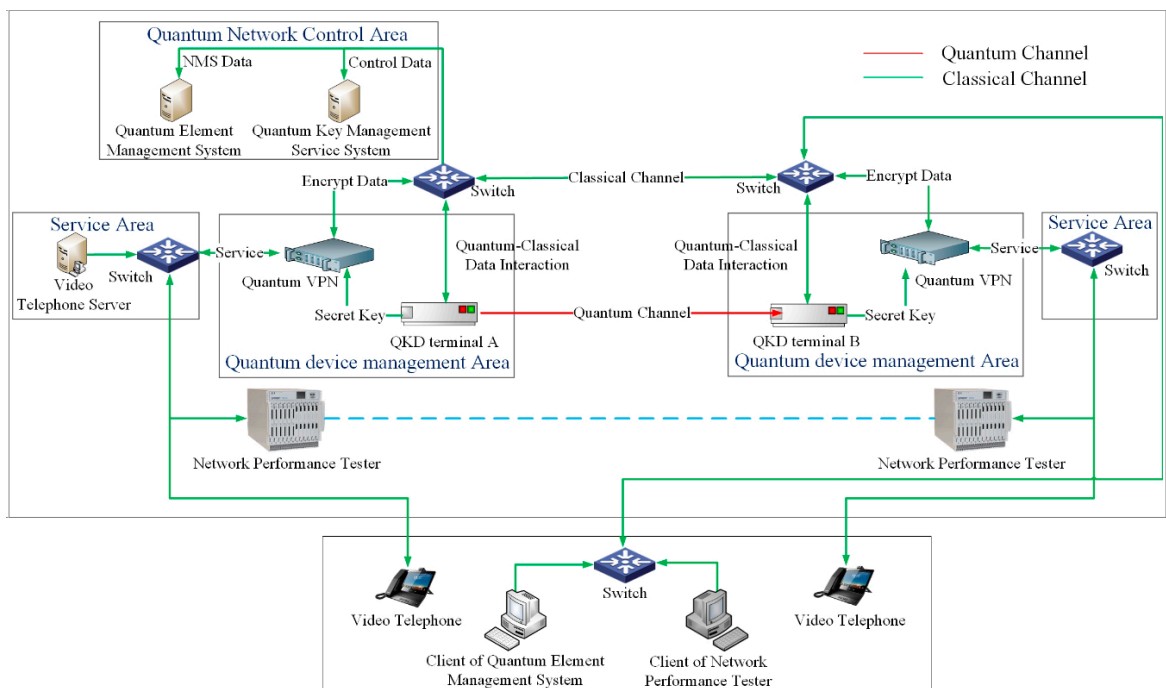

**Figure 3.** The network topology of the test environment.

**Table 3.** Experimental equipment.

| Name of Equipment | Model of Equipment |
| --- | --- |
| QKD terminal A [†] | QKDM-POL40A-24G4 |
| QKD terminal B [†] | QKDM-POL40B-24G4 |
| Quantum key management service system | NF-5270 M3 |
| Quantum network management system | NF-5270 M3 |
| Video telephone server | eSpace U1910 |
| Quantum VPN | SJJ1529 IPsec VPN-Q |
| Switch | s5700 |
| Video telephone | espace8950 |
| Network performance tester | N11U |
| Optical time-domain reflectometer | FTB-200 |
| Optical fiber power meter | JW3116/JW3216A |
| Digital anemometer | HT-628 |
| Optical fiber tray | 2# (10 km), 3# (10 km), 6# (20 km) |
| Fused fiber | 1*, 2*, 3* |

[†] QKD terminal is introduced at http://www.quantum-info.com/English/product/2017/0901/324.html.

Table 4 shows some experimental results of distance loss test and gallop loss test. As the length of quantum channel increases, the line distance loss increases, and the quantum key rate decreases. The secret key rate of the quantum device fluctuates greatly in the light breeze and gentle breeze. The moderate breeze blows up the optical fiber to reduce the swing amplitude and increase the secret key rate. The 40 km optical fiber is affected by the dual effects of line attenuation and swing, and the secret key rate drops sharply. Based on the analysis of results, the distance loss greater than 6 dB causes the secret key rate of the overhead line to drop sharply. When there is no wind disturbance,

the quantum key generation terminal meets the code limit (average secret key rate ≥ 2 Kbps). When the line is disturbed by wind, the quantum key generation terminal cannot be coded. Therefore, the anti-interference ability of the equipment and the application of quantum relay technology need to be considered in actual engineering.

**Table 4.** Quantum key rates of distance loss test and gallop loss test.

| Distance (km) | Loss (dB) | Average Secret Key Rate (Kbps) | | | | |
|---|---|---|---|---|---|---|
| | | Calm | Light Air | Light Breeze | Gentle Breeze | Moderate Breeze |
| 10 | 1.86 | 40.75 | 39.55 | 37.50 | 38.77 | 38.68 |
| 20 | 3.88 | 32.64 | 31.81 | 30.48 | 29.09 | 29.66 |
| 30 | 5.98 | 26.03 | 20.65 | 20.12 | 21.28 | 21.90 |
| 40 | 7.65 | 5.56 | 0.80 | 0 | 0 | 0 |
| 16.58 | 8.71 | 6.24 | - | - | - | - |

The connection loss is not only affected by the splicing of different spliced optical fibers, but also by the number of spliced nodes. The increase in the number of nodes leads to an increase in the degree of light scattering during transmission, and the connection loss also increases. As shown in Figure 4, compared with 2#, 3# and 6# fiber optic trays, the use of 1*, 2* and 3* spliced fibers increased the losses by 1.243 dB, 1.566 dB and 1.897 dB, respectively. Meanwhile, when the 2# and 3# fiber optic trays are connected, the macro-bend radius of less than 4 cm will cause the line to fail to communicate properly. From Table 5, it can be concluded that the average secret key rate fluctuates greatly with little change in attenuation. It is demonstrated on the side that the instability of wind force has a greater impact on the key rate of quantum overhead lines. At the same time, the average secret key rate of occasions with less spliced optical fibers was reduced by 11.285 Kbps, 13.103 Kbps and 1.489 Kbps, respectively.

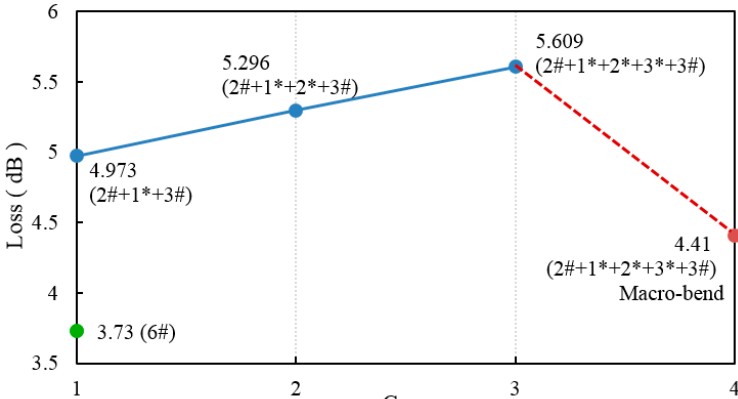

**Figure 4.** Test of connection loss (2# + 3# and 6#).

**Table 5.** Connection loss based on light breeze.

| Test Case | Loss (dB) | Average Secret Key Rate (Kbps) |
|---|---|---|
| 6# + 1* | 3.736 | 20.194 |
| 6# + 2* | 3.738 | 18.374 |
| 6# + 3* | 3.737 | 29.990 |

Figure 5 describes the experimental results of QVPN unidirectional transmission throughput under different encryption methods. In addition to the byte size of 1280 B, the "AES + quantum key" obtains the best performance among three algorithms. By analyzing the test results, the QVPN has insufficient support for AES algorithm when the packet byte size is 1280 B. When the byte size is 64 B, the "SM4 + quantum key" achieves the same throughput performance as the IKE algorithm.

In the case of other byte sizes, the throughput performance of the SM4 algorithm is lower than the IKE algorithm. Since the QVPN uses software technology to implement the SM4 algorithm, as an algorithm of the same order of magnitude as AES algorithm fails to achieve better performance, we should find solutions from the software and hardware collaborative architecture to improve the performance of QVPN for SM4 algorithm.

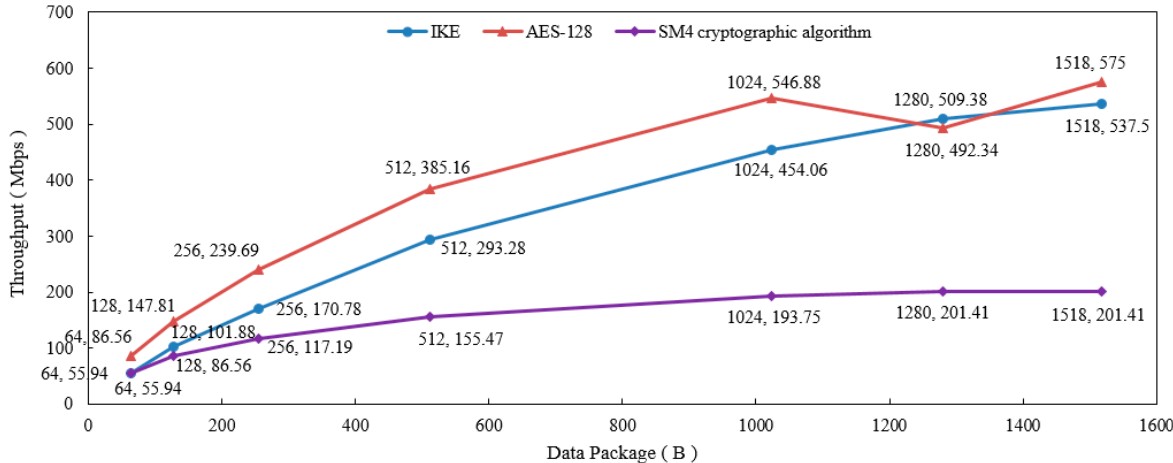

**Figure 5.** Comparison of unidirectional transmission throughputs among three algorithms.

QVPN provides quantum keys for SM4 algorithm and AES algorithm to encrypt and decrypt data. SM4 algorithm has a greater impact on data transmission performance than AES algorithm, and the performance degradation range is about 40%–50%. Meanwhile, the network performance tester was used to test business data traffic of different scales. The results show that when the IKE algorithm is used, the maximum throughput of QVPN unidirectional transmission is 538 Mbps. Under the condition of the SM4 algorithm, the maximum unidirectional throughput supported by the QVPN is 201 Mbps. For AES algorithm, the maximum unidirectional throughput supported by QVPN is 575 Mbps. QVPN can stably transmit without exceeding the maximum throughput of each encryption algorithm. Table 6 shows the stability test results of the secure transmission system for QKD. In the actual environment, the quantum equipment can be stable and coded for a long time, the control system runs smoothly, and it can be confirmed that the QKD system can run stably for a long time.

**Table 6.** Stability test.

| Test Case | Environment | Average Secret Key Rate (Kbps) |
|---|---|---|
| 1st day | | 21.22 |
| 2nd day | | 22.39 |
| 3rd day | 16.58 km | 22.28 |
| 4th day | 8.72 dB | 23.46 |
| 5th day | Light Breeze | 20.28 |
| 6th day | | 20.28 |
| 7th day | | 22.02 |

According to the analysis of the above test results, it can be seen that gallop loss and connection loss are the key factors that affect the performance of the key layer QKD. At the business layer, the choice of encryption algorithm can affect the interactive performance of power services. Combined with the test results under power overhead cable environment, a fast polarization feedback algorithm has been proposed to improve the performance of QKD [29,30]. This algorithm utilizes wavelength division multiplexing technology to achieve both strong and weak light fiber transmission. At the receiving end, the light is broken down to detect polarization changes in the photon state of the intense light. Then, by establishing a corresponding change model, the polarization state of the weak light is

corrected using an electric polarization controller. In addition, it solves the problem that it is difficult to encode due to frequent interference of photon polarization states. For the practical application of power QKD technology, the accuracy of fiber optic splicing needs to be improved. It is also necessary to reduce the number of fiber channel connections. Taking into account the different operating environments of various business systems, selected equipment and business requirements, encryption algorithms should be selected according to the needs of the actual business, and an efficient power QKD network with multiple encryption algorithms co-existing should be constructed.

## 4. Structure of Power QKD Network

The traditional power system constructs a network security protection system through security equipment such as firewalls, intrusion detection systems, VPNs and encryption devices, which ensures the secure transmission of data to a certain extent. However, new network attack methods are emerging in an endless stream, which poses new challenges to the security, reliability and timeliness of control/management instruction transmission. In practice, some businesses rely on offline keys to update keys, and the update frequency is slow, which affects the security level of information transmission. Considering that the traditional key online transmission has the hidden danger of being stolen, resulting in insufficient security and timeliness of online key distribution, and further affecting the security performance of the power system. In view of the above, this section proposes a security protection mode based on power QKD technology. At the same time, a new generation of data security transmission architecture based on QKD for dispatching automation, distribution automation, electricity information collection and video conference are designed.

### 4.1. QKD-Based Power Dispatching Automation

The safe and stable operation of the power dispatching automation system needs to meet the characteristics of reliability, safety, integrity, real time and consistency, etc. It is necessary to ensure the safe transmission of instructions and improve the anti-intrusion ability of the system. Traditional power dispatching longitudinal encryption and authentication device (LEAD) has the problems of slow key update and incapable of online secure distribution [31]. Precise data encryption technology based on power QKD technology can effectively prevent threats such as tampering, theft and illegal injection during the transmission of dispatching instructions. Figure 6 shows the QKD-based dispatching automation encryption transmission architecture. In this paper, QKD device is used for key negotiation to avoid key online transmission, and the key online real-time update of LEAD is realized, which solves the shortage of online distribution and update timeliness of traditional encryption key. By adding a key acquisition module, LEAD is transformed, and the method of obtaining the key by encryption device is changed to obtain the quantum key from QKD device. It realizes data encrypted transmission based on quantum key, such as telemetry, remote signaling, remote adjustment, etc. In addition, it guarantees the safe delivery of remote adjustment and remote-control information, the secure uploading of remote monitoring and remote signaling information of dispatching substations, and thus, the secure transmission of power dispatching information.

### 4.2. QKD-Based Distribution Automation

Distribution automation is an important technology that serves the construction of urban distribution networks. The safety of distribution instructions is critical. Compared with dispatching automation, the distribution network is multi-faceted, multiple communication methods coexist, and are greatly affected by environmental and social factors. It is prone to the risk of theft or tampering of unidirectional authentication and message transmission due to insufficient border protection [32]. Considering that the information exchange of new-type urban distribution networks is more frequent and more open, the use of QKD technology to encrypt the transmission data of distribution automation production is proposed. This study uses quantum keys to perform encryption on important authentication information and uses quantum relay technology to ensure long-distance

transmission of quantum networks. Finally, tamper-proof and anti-eavesdropping of information distribution is realized, thereby improving the anti-risk capability of the distribution network and ensuring the quality of electricity for urban residents. Figure 7 shows the network structure of quantum key encryption for distribution of automation data.

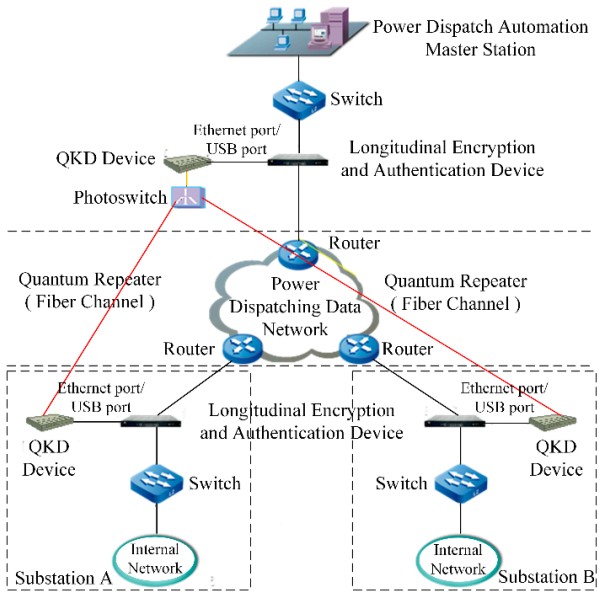

**Figure 6.** Dispatching automation system based on QKD.

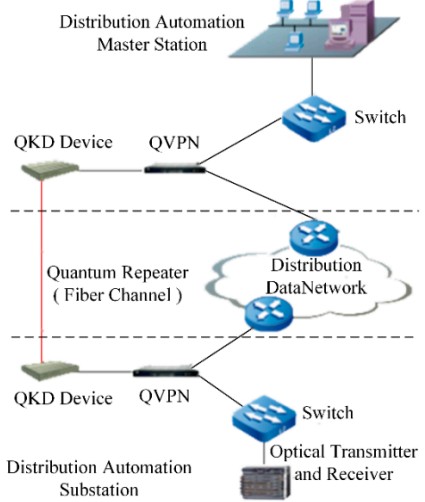

**Figure 7.** Distribution automation architecture based on QKD.

### 4.3. QKD-Based Electricity Information Collection

The electricity information collection system is one of the important components of the smart grid. It provides timely, complete and accurate feedback of user power consumption information to provide a reference for power control. Since the power consumption information contains a large amount of private data, if the data are manipulated by a third party, it will not only affect the daily power consumption of users, but also cause the leakage of sensitive data and private information [33]. In order to ensure the safe collection of a large number of electricity consumption information, it is necessary to build a highly secure and reliable transmission network. Therefore, QKD technology is used to increase the update frequency of the encryption key for the electricity information collection

service, and to ensure the secure data transmission of the electricity information collection system. Figure 8 shows the structure of QKD-based electricity information collection.

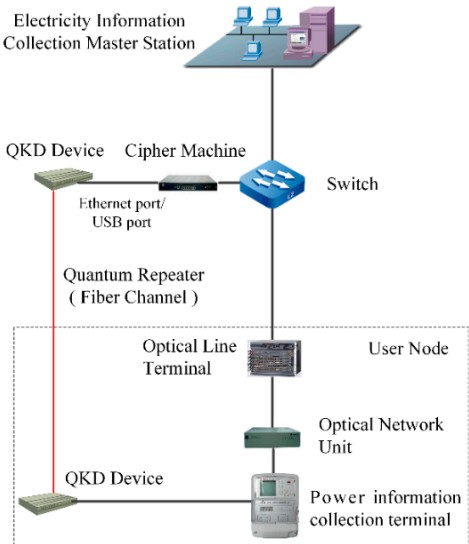

**Figure 8.** Electricity information collection based on QKD.

### 4.4. QKD-Based Video Conference System

The large-scale popularity of the video conference system has improved the efficiency of administrative negotiations, cross-regional consultations and cross-regional exchanges, while reducing the costs of meetings. Accompanying system security issues have gradually become apparent and major decision-making consultations face third-party threats. At the same time, video information has the characteristics of large data volume, controllable encoding rate and strict synchronization. The encryption method is different from text encryption. Combined with the complicated overhead power lines in operation, in order to guarantee the performance of quantum key distribution, QKD equipment with fast deviation feedback function is equipped, and quantum relay equipment is used to guarantee the key rate on large-span lines. As a result, it satisfies the need for secure transmission of large amounts of video data. Finally, high-efficiency, high-security data protection is achieved, and the level of security protection for video conference scenarios in fixed venues and outdoor emergency is improved. Figure 9 shows the operation chart of video conference.

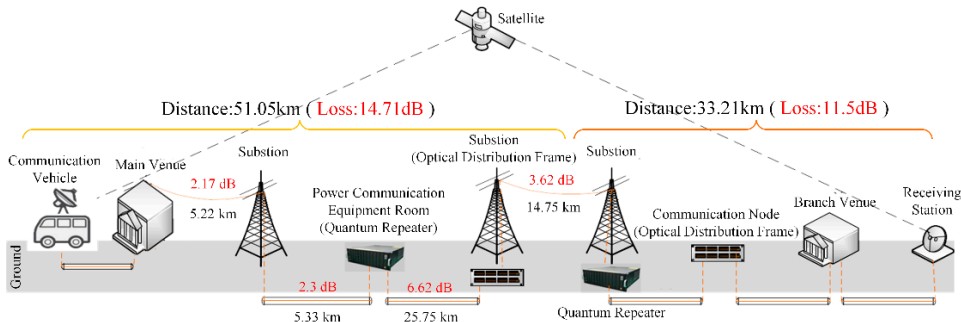

**Figure 9.** Video conference system based on QKD.

### 4.5. QKD-Based Data Disaster Recovery System

Data disaster recovery refers to the establishment of a remote multi-point backup of data through the establishment of a disaster recovery center to improve resistance to operations during a disaster. According to the security level protection and grading standards of power companies, the security

protection level of disaster recovery and backup services is S2A2G2 in the second level. The protection measure of traditional system is to deploy servers with different risk levels in different physical areas, and each area is safely isolated by a firewall. The protective measures are limited, and the level of protection is insufficient. Since the disaster recovery data involves sensitive information of most of the core power business, ensuring the safe transmission of core data of the power grid business is a point worth paying attention to in disaster recovery backup. In order to improve the security of disaster recovery services and rely on the secure distribution mechanism of quantum keys, this paper designs a disaster recovery service system based on the encryption and decryption of quantum key. Figure 10 shows a QKD-based disaster recovery backup system.

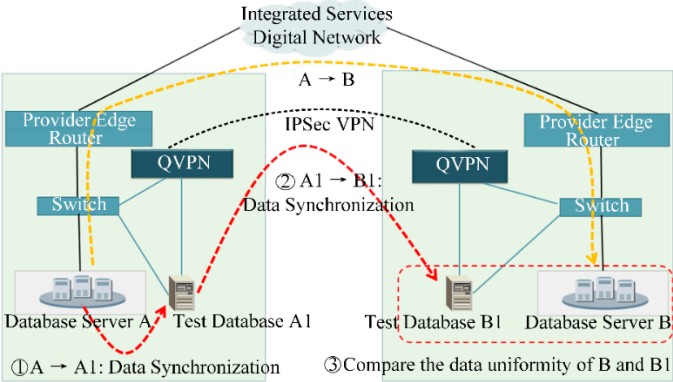

**Figure 10.** Disaster recovery backup based on QKD.

## 5. Experimental Results

Relying on the existing power network, this experiment established a verification environment by using QKD equipment, QVPNs and qualified fiber channels. Note that, actual business characteristics and information security requirements were considered. The test selects actual links to conduct business tests of dispatch automation, video consultation and data disaster recovery. Furthermore, the business performance tests of power distribution automation and power information collection are conducted in a simulated environment.

As shown in Figure 11, the total length of the test line for the power dispatch automation industry was 142.82 km, with the measured attenuation of 53.76 dB. There are three types of the line environment used: BOFC, all-dielectric self-supporting optic fiber cable and OPGW. Due to the aging of the lines, the length and attenuation of BOFC was 43.94 km and 22.49 dB, respectively. The aerial part extends for 98.88 km, with the attenuation of 31.3 dB. The key negotiation between transmitter and receiver based on decoy-state BB84 QKD protocol [29]. The wavelength of quantum signals was 1550 nm. The transmitter and receiver are located in Headquarter A and substation G, respectively. Headquarter A was the National Power Dispatch Control Center (NPDCC) node and it was the dispatching automation master station. Node B was the centralized control node. Provincial company C was the Provincial Power Dispatching Control Center (PPDCC) point and it was the dispatching automation substation. Substation G was the power plant station. Headquarters node A, provincial node C and substation G form a three-level scheduling structure. A 2 Mbit/s channel was built through synchronous digital hierarchy (SDH) optical transmission system to realize the point-to-point interconnection of QVPN gateway of "dispatching master station to substation" and "substation to plant station" links. From substation G to provincial node C, there are three relay nodes, substation D, substation E and substation F. the whole network was composed of three levels of seven nodes. The test connects the remote host at the plant station end to the QVPN gateway for business traffic encryption and tunnels through the 2 Mbit/s channel. After reaching the remote QVPN gateway, the remote QVPN gateway decrypts it and forwards it to the target device to complete the transmission of QKD data for dispatching automation services.

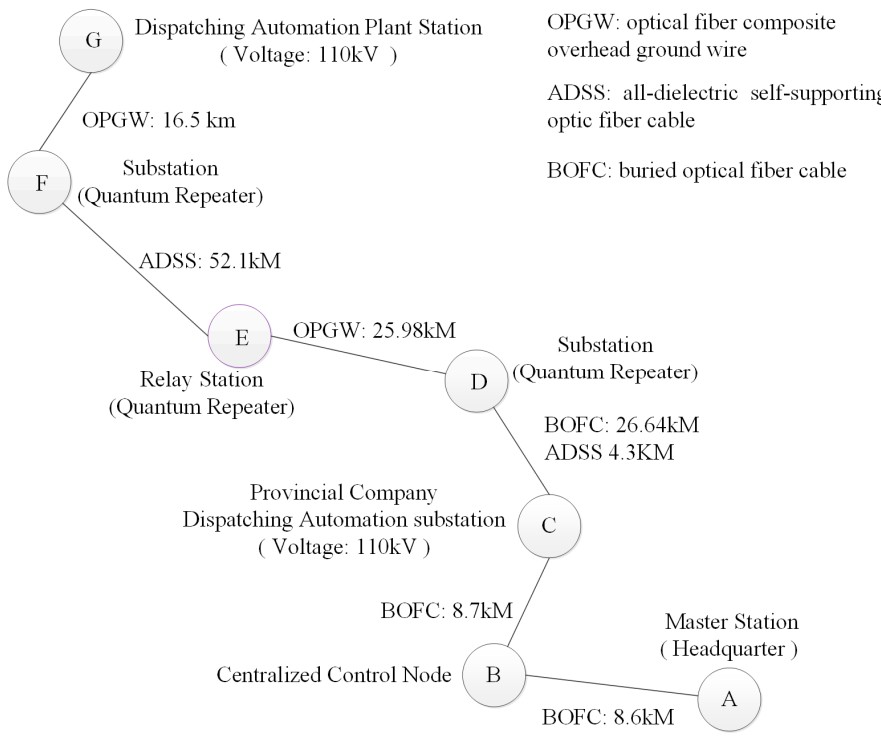

**Figure 11.** The actual line of dispatching automation test.

The purpose of this test was to verify the performance of the QKD system for encrypted transmission of business instructions such as remote signaling, telemetry, remote control and remote adjustment. In the test, the main station, substation and remote control device were deployed at the National Power Dispatch Control Center point A, the Provincial Power Dispatching Control Center point C and power plant station point G and the terminal professional software was used to simulate the protection device to connect with the remote control device. The dispatching communication network port was connected to a quantum encryption VPN device to encrypt service data streams. The verification of the remote signaling, telemetry, remote control and remote adjustment services of the protection device and the main station/substation end was carried out. The telemetry service data at the plant end was taken from the real telemetry data in the station.

Figure 12 shows the diagram of QKD-based dispatching automation test. The RTU (original production system) sends the remote signal and telemetry information in the station to the provincial dispatching automation system and the mobile master station (substation). Then, the telemetry and telemetry information received by the dispatching automation system and mobile master station (substation) are compared for consistency. The plant station G sends remote control and remote adjustment instructions through the RTU, analyzes the messages received by the RTU and performs consistency comparison with the messages sent by the mobile master station (substation). This experiment applied QKD to remote signaling, telemetry, remote control and remote adjustment services. Experimental results show that QKD-based dispatching automation obtains the consistency of four business interaction information between plant station G and master station A. Meanwhile, the dispatching automation system works normally according to the received instructions, and the transmission delay was less than 3 ms, which meets the indicators of stable operation of power control businesses. Relying on the test environment, this experiment verified the performance of the fast polarization feedback algorithm. Under the condition that the mean value of line polarization change rate was less than 6.27 rad/s, the average time for the single quantum state correction of QKD equipment based on fast polarization feedback technology was less than 6.95 ms (test time period: 10 min). In the experiment, a standard single-mode fiber (G652, loss: 15 dB) with a length of 65.7 km was used to test the key rate of the QKD system. The average secret key rate was about 3.8 Kbps.

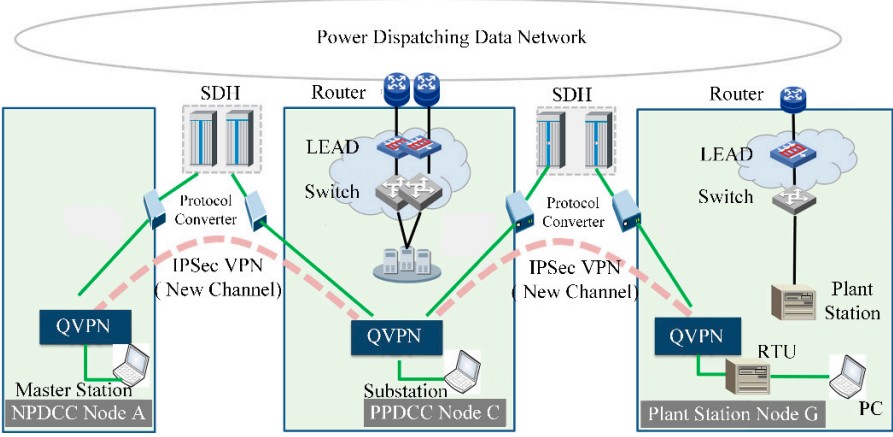

**Figure 12.** Schematic diagram of QKD-based dispatching automation test.

Considering that the video conference was affected by multiple factors, such as the environment, lines and equipment, this paper implements a multi-relay hybrid networking application in the video conference in the power industry. As shown in Figure 9, the demonstration project spans three substations and two-segment overhead optical cables of a multi-relay power QKD network, solving the technical problem of stable quantum key coding in a complex power communication network environment. At the same time, we applied QKD to satellite emergency communication systems, to achieve the security of wireless data transmission. Quantum fast polarization feedback technology was used to overcome the influence of the line environment on the quantum key coding. A correction model was built by quickly detecting changes in the quantum polarization state in an overhead power cable. The correction of abnormal changes was completed, which effectively improved the adaptability of QKD technology in complex power network environments. Based on the fast polarization feedback technology, the actual average effective coding time of the quantum channel was increased from 68.51% of ordinary devices to 99.76%, and the actual daily average effective secret key rate was increased from 63.41% of ordinary devices to 83.51%.

As shown in Figure 11, in this QKD-based disaster recovery service test, the actual line between the main station A and the substation C was selected to set up a verification environment for QVPN encrypted transmission. The main station A and the substation C are the clients and servers of the disaster recovery service test, respectively. By using the FTP data download method, the QKD secret key rate and QVPN transmission performance were tested. When the line attenuation of the verification environment was 13 dB, the average secret key rate of the quantum key can reach 27.79 Kbps. When the service packet length was greater than 1024 bytes, the QVPN transmission rate was 581 Mbit/s (the time delay was less than 8 ms) and the database backup comparison was consistent, which meets the transmission requirements of power disaster recovery services. At the same time, this experiment set up a QKD-based distribution automation and power information collection service test environment, in which the bandwidth was 2 Mbit/s, the encryption algorithm was SM4 and the update frequency of the quantum key was 16 times/s. The test results show that the delays of the proposed scheme are 3.67 ms and 3.68 ms, respectively, which meets the delay standard requirements of service operation and the accuracy requirements of data transmission. Note that, for the delay standard requirements, the distribution automation business (control area, one-way) has a latency of less than 500 ms, and the latency of the power information collection service was less than 60 s.

In summary, based on the core business test results of power QKD technology, the proposed scheme meets various indicators for stable transmission of the power grid. It can effectively ensure the security of energy information interaction and further strengthen the support for intelligence and digitalization of power systems.

## 6. Conclusions

In view of the complexity of QKD technology in power grid applications, the feasibility of the system was analyzed, from the performance parameters related to the key tier and business tier of the power QKD system. We have proposed a performance evaluation framework from six aspects. The performance simulation test data of quantum key rate were tested and evaluated. On this basis, this study builds a QKD network suitable for different power businesses, based on actual power lines. Moreover, we have carried out feasibility verifications of a QKD networking solution in combination with production business. Meanwhile, a fast polarization feedback algorithm was used to overcome the impact of actual overhead line dancing on key rates. Various business indicators meet actual power production business needs. Among them, the shorter the quantum signal state correction time, the higher the quantum key rate efficiency. Finally, our experiments not only confirm the effectiveness of QKD technology in power environment, but also provide a reference for the practical use of the technology in power.

**Author Contributions:** Conceptualization, B.Z.; methodology, X.Z.; data curation, Z.C. and R.S.; supervision, D.W.; investigation, T.P.; project administration, L.Y. All authors have read and agreed to the published version of the manuscript.

**Funding:** This work was supported in the National Key R & D Program of China (No. 2018YFB0805005).

**Conflicts of Interest:** The authors declare no conflict of interest.

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
