# Peer review of "Performance Analysis of Quantum Key Distribution Technology for Power Business"

_applsci, doi:10.3390/app10082906_

Round 1

Reviewer 1 Report

  1. There are several statements in this paper which are left uncited. A few are indicated here, but there are others that the authors need to think about.
    1. Lines 35 – 42. There are statements on analysing attacks on systems, but no citations, as a reader outside the attack area I don’t know what you are trying to prevent.
    2. Lines 122 – 124. QKD protocols stated but uncited. Readers interested in looking further into the topic don’t know where to find more details
    3. Line 178 – the QKD terminal models. What company are they from? A citation to the company providing these terminals would be useful, if available.
  2. Use of QSC and QKD – QSC encompasses a wide range of other quantum communication protocols beyond QKD. For instance, protocols like quantum digital signatures, quantum fingerprinting, and quantum oblivious transfer. The term is used almost interchangeably in the paper. My advice would be to highlight QSC as the overarching interest, then specify the paper here only considers QKD, and use QKD from then on.
  3. Acronyms in general – acronyms in diagrams need to be defined in the text and in the captions. There are lots of acronyms in this paper, making it hard to keep track of each, especially when they aren’t in figure captions.
  4. Paper structure – the paper reads as two papers sandwiched together. If the article was presented as section 1, 2, and 3 with a conclusion, or as sections 1, 2, 4, 5, and a conclusion, it would be a more effective paper. The authors could present two papers with more experimental details and methods.
  5. Paper structure - is section 2 needed? The explanations can be found in the literature.
  6. What is the reason behind the polarisation-based protocol for optical-fibre implementation?
  7. Experimental details – the experimental information and methods are generally not there. As a QKD reader, I would want to know more about the protocol implemented and the architecture and performance of the QKD transmitter and receiver, which are left out.
  8. Experimental results – non-conventional terms used for results, such as “code rate” and “average qubit rate”. Raw key rate, sifted key rate, and secure key rate are required for impact in the field.
  9. Where are the quantum bit error rate results? These could be used to explain results from line 236 to 238, where the attenuation is the same, but code rate (secure key rate?) fluctuated.
  10. Lines 118 – the different types of QKD list is incomplete. QKD protocols include weak-coherent pulses, true single-photon signals, entangled photon signals, and continuous variable. Line 119 states that single-photon signals are more practical; however, line 125 states that single-photon sources are difficult due to technology. What the author means is the weak-coherent pulse protocols are easy to implement.

Reviewer 2 Report

Please refer to the review comments below.

1. It is need to clarify the subject of this paper.
It is not clear whether it is a study of QSC communication or a technical review of how QSC communication is applied to the power business.

2. If the focus is on the QSC communication method, the contribution of the author in the QSC communication method study should be clearly stated.

3. If the purpose of this paper is introducing how to apply the QSC communication method to the power business then it is need to focus on the specificity of the power business and the effect of applying the QSC communication method.

4. In conclusion, the content is generally lengthy and far from the general method of writing  a journal paper.

Reviewer 3 Report

  1. The abstract should be rewritten to highlight the motivation of the paper.
  2. The introduction part has to be more organized and updated with recent citations.
  3. The simulation results have to be explained in details.
  4. The authors have to check the format of references, table titles and figures for the whole paper.
  5. The conclusion part should be rewritten as a paragraph and not points.
  6. The whole paper has to check for sentence structure, flow of the paper, grammar and meaning.
  7. The term of QSC is confusing in the paper since there are many types of it like direct and deterministic.
  8. Current IoT devices are not capable of processing quantum stuff, How IoT devices will be contacted with Quantum Devices?
  9. Where the final key will be stored?
  10. The used layering approach is already there for a long time since DARPA started working on quantum communication and quantum cryptography.
  11. The authors didn't mention any information about the stages of implementing QKD and consider them for the simulation process.
  12. Simulation using a quantum internet simulator will be better

Round 2

Reviewer 1 Report

Dear Authors,

Thank you for responding to the comments raised in the first iteration. Your changes have clarified a number of aspects of the paper that were unclear in the first iteration. The slight restructuring has also made it more inviting to read.

Corrections recommended before publication:

Point 1 – Don’t use the “, etc.” in the abstract. Finish the example list after slice loss.

Reviewer 2 Report

None.
